# Toward Principled Transformers for Knowledge Tracing

## Abstract

Knowledge tracing aims to reason about changes in students' knowledge and to predict students' performance in educational learning settings. We propose knowledge tracing set transformers (KTSTs), a straightforward model class for knowledge tracing prediction tasks. This model class is conceptually simpler than previous state-of-the-art approaches, which are overly complex due to domain-inspired components, and which are in part based on suboptimal design choices and flawed evaluation. In contrast, for KTSTs we propose principled set representations of student interactions and a simplified variant of learnable modification of attention matrices for positional information in a student's learning history. While being largely domain-agnostic, the proposed model class thus accounts for characteristic traits of knowledge tracing tasks. In extensive empirical experiments on standardized benchmark datasets, KTSTs establish new state-of-the-art performance.

## 1 Introduction

The ultimate goal of knowledge tracing is to support students' learning processes (Corbett & Anderson, 1994). Knowledge tracing enables the adaptation of learning materials to students' individual needs and therefore constitutes an integral component of intelligent tutoring systems. In this paper, we focus on predictive performance and study knowledge tracing as a supervised sequential learning task in which we aim to predict the correctness of the next response of a student, given the history of her interactions with a learning system (cf. Gervet et al., 2020).

To achieve good predictive performance, existing approaches often rely on domain knowledge either regarding representations of students' interactions (Ghosh et al., 2020; Shen et al., 2022; Yin et al., 2023) or regarding model architectures (Zhang et al., 2017; Nakagawa et al., 2019; Long et al., 2021; Shen et al., 2021). So far, simpler models (Piech et al., 2015; Liu et al., 2023b) do not achieve comparable results on benchmark tasks and consequentially, the state-of-the-art in knowledge tracing prediction tasks consists of rather complex approaches. In addition, several recent approaches are based on suboptimal design choices and flawed evaluation. Specifically, they introduce an interaction representation that is inefficient, led to label leakage in the past, and introduces a distribution shift between training and evaluation (cf. Ghosh et al., 2020; Liu et al., 2022; 2023b; Yin et al., 2023).

We propose knowledge tracing set transformers (KTSTs), a straightforward, yet principled and performant model class for knowledge tracing prediction tasks, based on the standard transformer architecture (Vaswani et al., 2017). To account for characteristics of knowledge tracing tasks, we propose a simplified variant of learnable modification of attention matrices (cf. Press et al., 2021), as well as principled interaction representations that do not rely on domain-knowledge. Compared to flawed interaction representations in related work, our representations accurately reflect the learning setting and satisfy permutation invariance (cf. Zaheer et al., 2017) with regard to sets of input features. We explicitly discuss limitations of prior work in the problem setting (Section 3) and in contrast to KTSTs (Section 4). Empirically, we evaluate our method on eight datasets and compare it against 22 baseline models (Section 5), establishing new state-of-the-art performance on knowledge tracing benchmark tasks.

## 2 RELATED WORK

*Knowledge tracing* as a problem setting was established by Anderson et al. (1990). Classical machine learning approaches for knowledge tracing include probabilistic graphical models (e.g. Corbett & Anderson, 1994; Käser et al., 2017) and factor analysis-based approaches (e.g. Cen et al., 2006; Pavlik et al., 2009; Vie & Kashima, 2019), where many contributions proposed models that explicitly build upon domain knowledge regarding the educational learning setting (e.g. Pardos & Heffernan, 2011; Yudelson et al., 2013; Khajah et al., 2014).

*Deep knowledge tracing*, that is, the use of deep learning (LeCun et al., 2015; Schmidhuber, 2015) for knowledge tracing, was introduced by Piech et al. (2015). Approaches can be differentiated by their dominant modeling choices: Several methods leverage recurrent neural networks (RNNs, Hochreiter & Schmidhuber, 1997) for the sequential prediction task (Piech et al., 2015; Yeung & Yeung, 2018; Nagatani et al., 2019; Lee & Yeung, 2019; Sonkar et al., 2020; Long et al., 2021; Shen et al., 2021; 2022; Liu et al., 2023a); others include memory-augmented components (Santoro et al., 2016; Graves et al., 2016) to explicitly represent students' knowledge states (Zhang et al., 2017; Abdelrahman & Wang, 2019) or utilize graph neural networks (GNNs Scarselli et al., 2009) to capture relations between students and questions (Nakagawa et al., 2019; Yang et al., 2021).

*Transformer based knowledge tracing* approaches are characterized by building upon the transformer architecture (Vaswani et al., 2017). As transformers are integral components of state-of-the-art models for natural language processing (NLP, e.g. Brown et al., 2020) and for structured data in general (e.g. Dosovitskiy et al., 2021), they constitute a promising modeling choice for deep knowledge tracing. Knowledge tracing set transformers (KTSTs), as proposed in this paper, fall into this category. Prior work investigated how changes regarding the architecture and flow of information (Pandey & Karypis, 2019; Choi et al., 2020; Ghosh et al., 2020; Zhan et al., 2024), regarding positional information in the attention mechanism (Ghosh et al., 2020; Im et al., 2023), and regarding the interaction and knowledge component representation (Ghosh et al., 2020; Liu et al., 2023b; Yin et al., 2023) influence the predictive performance. In contrast to KTSTs, most related work include domain-inspired components that increase model complexity. Approaches that combine transformer based knowledge tracing architectures with other deep learning paradigms, such as hypergraph convolutions and RNNs (e.g. He et al., 2024), result in even more complex models. Self-supervised approaches for transformer based knowledge tracing (e.g. Lee et al., 2022; Yin et al., 2023) are orthogonal to our work.

We continue to discuss related work throughout the remainder of this paper: We address the short-comings of the prevalent, yet flawed, *expanded representation* for interaction sequences (Section 3). We point out domain-inspired components of previously proposed architectures in comparison to the straightforward architecture of KTSTs (Section 4.1). We compare attention mechanisms from related work in relation to the proposed learnable modification of attention matrices (Section 4.2). We contrast domain-inspired interaction representations for knowledge tracing with KTSTs' principled set representations regarding complexity and permutation invariance (Section 4.3).

## 3 PROBLEM SETTING

We study knowledge tracing as a supervised sequential learning task, that is, we predict the correctness of the next response by a student, given her learning history in form of a sequence of interactions with a learning system, where an interaction comprises all available information at a given time step (cf. Gervet et al., 2020). The prediction task can be formalized as follows. Consider a *sequence of interactions* of a student with a learning system. At any time $1 \leq t \leq T$, the student attempts to solve a *question* $\mathbf{q}_t \in \mathcal{Q}$ and her binary *response* $\mathbf{r}_t \in \{0, 1\}$ is observed, where $\mathbf{r}_t = 1$ indicates a correct and $\mathbf{r}_t = 0$ indicates an incorrect answer. Every question $\mathbf{q}_t$ is associated with one or more *knowledge components* $c \in \mathcal{C} = \{c_n\}_{n=1}^{|\mathcal{C}|}$ given by $\mathbf{c}(\mathbf{q}_t) \subseteq \mathcal{C}$. In this context, knowledge components describe information and skills that are required to solve specific tasks or questions as part of a domain model. We summarize the sequence of interactions of a student by $\mathbf{y}_{1:T}$ with $\mathbf{y}_t = (\mathbf{q}_t, \mathbf{c}(\mathbf{q}_t), \mathbf{r}_t)$. At time $t$, the machine learning task is to predict the next response $\mathbf{r}_{t+1}$ given the interactions $\mathbf{y}_{1:t}$ and the next question $\mathbf{x}_{t+1} = (\mathbf{q}_{t+1}, \mathbf{c}(\mathbf{q}_{t+1}))$.[1] Figure 1 visualizes the setting.

---

[1] We use $\mathbf{y}_t$ and $\mathbf{x}_t$ to denote complete interactions and questions, respectively. This allows us to provide a more concise description of the transformer architecture in Section 4.

**Limitations in related work**  Well published recent knowledge tracing approaches (e.g. Ghosh et al., 2020; Liu et al., 2022; 2023b) propose an interaction representation that stresses the effect of individual knowledge components on the learning task and that cannot properly handle multiple knowledge components per question. We refer to this representation as *expanded representation* (cf. Liu et al., 2022). In the *expanded representation*, interactions with multiple knowledge components per question are duplicated for every knowledge component, such that interactions involve only a single question associated with only a single knowledge component each. Suppose an interaction $\mathbf{y}_t$ is associated with multiple knowledge components $|\mathbf{c}(\mathbf{q}_t)| \geq 2$. Its *expanded representation* is $\tilde{\mathbf{y}}_t = (\mathbf{q}_t, \mathbf{c}(\mathbf{q}_t)_1, \mathbf{r}_t), \ldots, (\mathbf{q}_t, \mathbf{c}(\mathbf{q}_t)_{|\mathbf{c}(\mathbf{q}_t)|}, \mathbf{r}_t)$, resulting in a new interaction sequence of length $\tilde{T} \geq T$. One consequence is an increase in sequence length by a factor depending on the average number of knowledge components per question. In prior work, models are trained to predict the next response $\mathbf{r}_{\tilde{t}+1}$ for $1 \leq \tilde{t} \leq \tilde{T}$, based on this expanded representation. Without proper masking

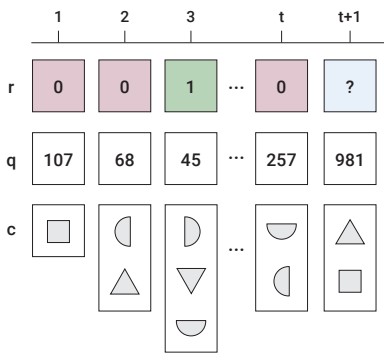

Figure 1: Knowledge tracing as supervised sequential learning task: predicting the next response, given a history of questions, knowledge components, and previous responses. Questions are represented by toy IDs, knowledge components are visualized as discrete shapes.

regarding the original learning task and interaction sequence, this introduces *label leakage* (e.g. in Ghosh et al., 2020; Yin et al., 2023). For inference, label leakage has been fixed in Liu et al. (2022), where the prediction for original response $\mathbf{r}_t$ is computed by aggregating individual predictions based on $\tilde{\mathbf{x}}_t$, the expanded representation of question $\mathbf{x}_t$. Given this fix, however, the underlying distribution at inference time differs from the training distribution, where the distribution shift results in suboptimal predictions. This effect is pronounced when models improperly learn to rely on label leakage during training. Empirically, we find that the performance of the expanded representation is noticeably worse for datasets with knowledge-component-to-question ratio larger than two (Section 5.1). We address the flaws of the expanded representation in our contribution, introduced in the following section.

## 4 KNOWLEDGE TRACING SET TRANSFORMERS

In this section, we propose sequential Knowledge Tracing Set Transformers (KTST). Our approach is based on a standard transformer architecture (Vaswani et al., 2017) with adjustments for best performance on knowledge tracing tasks. It features a simple, yet learnable variant of modification of attention matrices for positional information (cf. Press et al., 2021) and operates on principled representations of interactions that do not rely on domain-knowledge and which are permutation invariant (cf. Zaheer et al., 2017) with respect to sets of knowledge components.

### 4.1 TRANSFORMER ARCHITECTURE FOR KNOWLEDGE TRACING

The deep learning architecture for KTSTs is based on the standard transformer architecture (Vaswani et al., 2017) including an encoder and a decoder, multiple layers with multi-head self-attention (MHSA) and cross-attention, residual connections (He et al., 2016), dropout (Srivastava et al., 2014), layer normalization (Ba et al., 2016) and feed-forward neural networks. Instead of positional encodings, we propose to use a learnable modification of attention matrices to inform the model about positional information (see Section 4.2); other differences are pointed out in the following. Overall, the architecture allows for efficient training with a single forward and backward pass per sequence.

For knowledge tracing prediction tasks (see Section 3), KTSTs operate on multiple discrete tokens per interaction $\mathbf{y}_t$: a question $\mathbf{q}_t$, its associated knowledge components $\mathbf{c}(\mathbf{q}_t)$, and response $\mathbf{r}_t$. We embed every token into a vector of size $d$. Embeddings associated with the same time step are aggregated to form a joint interaction representation (respectively question representation, see Section 4.3). For each interaction, KTSTs estimate the probability of a correct response at the next time step, $P(\mathbf{r}_{t+1} = 1 | \mathbf{x}_{t+1}, \mathbf{y}_{1:t})$, where the representation of the next question $\mathbf{x}_{t+1}$ is used as *query token* in the decoder, attending to the learning history $\mathbf{y}_{1:t}$ as processed by the encoder. We

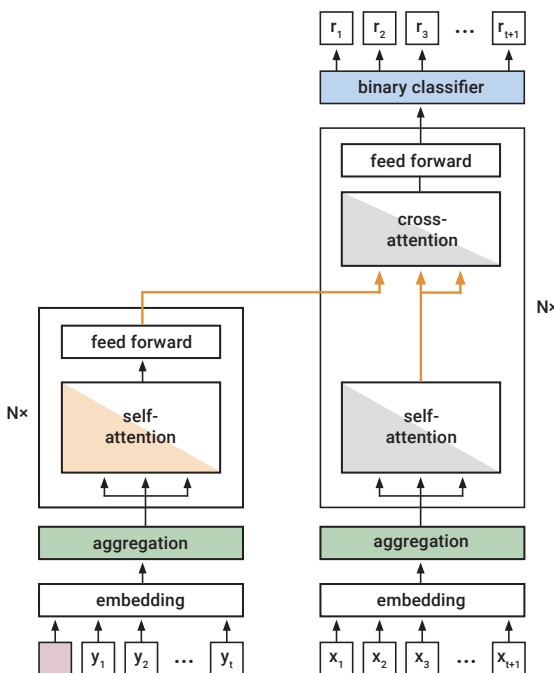

Figure 2: Overview of the transformer architecture used in KTSTs, where we abstract from skip connections, dropout, normalization and feed-forward layers. Differences from the standard transformer architecture are highlighted with coloring.

mask encoder and decoder appropriately. In the encoder, we use a triangular causal mask with the learning history shifted by one time step. Different from standard transformers, we use the encoded $\mathbf{y}_{1:t}$ only as *value* in the cross-attention layer, such that questions $\mathbf{x}_{t+1}$ serve as both *query* and *key* within the cross-attention. For knowledge tracing, this adjustment has been proposed by Pandey & Karypis (2019); within our architecture it is empirically supported by an ablation study (cf. Section 5.2). A binary classifier operating on the output of the decoder yields the probability estimates. See Figure 2 for a visualization.

**Comparison with domain-inspired architectures in related work**   The architecture proposed for KTSTs is technically straightforward and does not rely on domain-specific components. This is in contrast to many (transformer based) deep knowledge tracing architectures, which impose a strong inductive bias with respect to the task at hand and are often motivated by concepts associated with human learning. Examples of inductive biases in related work include: encoding students' knowledge states by means of RNN hidden states (Long et al., 2021), explicitly modeling a student's knowledge acquisition process with memory-augmented neural networks (Zhang et al., 2017; Abdelrahman & Wang, 2019), enforcing a smoothness constraint on learnable knowledge parameters which are assumed to represent knowledge components (Yin et al., 2023), explicitly including question difficulty estimates that guide updates within the architecture (Shen et al., 2022), and explicitly modeling the relationship between learners and questions via graph neural networks (Nakagawa et al., 2019; Yang et al., 2021). Intuitively, these components increase model complexity. Although KTSTs require only small changes to the standard transformer architecture, they outperform more complicated approaches empirically (see Section 5), and are conceptually simpler.

## 4.2 LEARNABLE MODIFICATION OF ATTENTION MATRICES

In transformer architectures, positional information is usually conveyed either by including positional embeddings or by modifying attention matrices (Dufter et al., 2022). For KTSTs, we propose a multi-head attention mechanism with learnable exponential decay applied to attention weights at previous time steps $<t$ to account for the sequentiality of the learning task following Press et al.

(2021). Exponential decay applied to attention weights reduces attention scores based on the relative distance of key tokens to the query token. Similar attention functions have, for example, been investigated for knowledge tracing tasks by Ghosh et al. (2020) and Im et al. (2023).

Let $\alpha_{t,t-\tau} \in [0,1]$ denote the attention weights calculated in a single head for *query* $q_t \in \mathbb{R}^d$ and *key* $k_{t-\tau} \in \mathbb{R}^d$ at time steps $t$ and $t-\tau$, respectively, with $0 \leq \tau \leq t-1$. Assuming causal masking, where attention weights for $\alpha_{t,>t}$ are set to 0, the *standard* scaled dot product attention between two tokens as introduced in Vaswani et al. (2017), results from the application of a softmax

$$\alpha_{t,t-\tau}^{\text{standard}} = \frac{\exp\left(e_{t,t-\tau}\right)}{\sum_{\bar{\tau}=0}^{t-1} \exp\left(e_{t,t-\bar{\tau}}\right)},$$

to (in general unbounded) attention scores $e_{t,t-\tau}$. Abstracting from in-projections, $e_{t,t-\tau}$ is given by

$$e_{t,t-\tau} = \frac{q_t^\top k_{t-\tau}}{\sqrt{d}}.$$

In KTSTs' attention mechanism, we subtract a positive value $\tau\theta \geq 0$ from attention scores $e_{t,t-\tau}$, resulting in

$$\alpha_{t,t-\tau}^{\text{KTST}} = \frac{\exp\left(e_{t,t-\tau} - \tau\theta\right)}{\sum_{\bar{\tau}=0}^{t-1} \exp\left(e_{t,t-\bar{\tau}} - \bar{\tau}\theta\right)} = \frac{\exp\left(e_{t,t-\tau}\right) \cdot \exp\left(-\tau\theta\right)}{\sum_{\bar{\tau}=0}^{t-1} \exp\left(e_{t,t-\bar{\tau}}\right) \cdot \exp\left(-\bar{\tau}\theta\right)}.$$

Here, $\theta$ is a learnable parameter per attention head which we restrict to be positive via application of the softplus function (Dugas et al., 2000; Glorot et al., 2011). We initialize effective values of $\theta$ according to the geometric series as prescribed by Press et al. (2021) for ALiBi (*attention with linear biases*, where *linearity* refers to attention scores). Hence, for models with four attention heads, $\theta$ is initialized with $\frac{1}{2^2}, \frac{1}{2^4}, \frac{1}{2^6}$ and $\frac{1}{2^8}$, respectively, while models with eight heads give rise to the initialization $\frac{1}{2^1}, \frac{1}{2^2}, \ldots, \frac{1}{2^8}$.

The initialization introduces an inductive bias at the beginning of the training process, where interactions with a higher relative distance are assigned a lower weight in the attention mechanism. In general, KTSTs' attention mechanism results in attention weights that are exponentially decayed. For $\theta = 0$, we have $\alpha_{t,t-\tau}^{\text{KTST}} = \alpha_{t,t-\tau}^{\text{standard}}$, whereas for large values of $\theta$, we have $\alpha_{t,t}^{\text{KTST}} \approx 1$ and $\alpha_{t,<t}^{\text{KTST}} \approx 0$. In absence of a positional encoding, this results in an attention function that interpolates between attention on a set for $\theta = 0$ and soft sliding windows (which are possibly very narrow) applied to interactions at time steps $<t$ for large $\theta$. Empirically, the proposed handling of positional information within the attention mechanism performs best for KTSTs (see Section 5.2).

**Modification of attention matrices in related work** In KTSTs we employ learnable modification of attention matrices for positional information as described above. We highlight how the mechanism has an inductive bias that puts more weight on recent interactions. In related work, Im et al. (2023) associate a decay in attention weights with students' *forgetting behavior*, while Ghosh et al. (2020) argue for a *context aware* modification of attention weights, that explicitly includes the similarity of knowledge components (in contrast, we argue that the latter is already captured within standard attention). The exponential decay in Im et al. (2023) is very related to our proposed approach (as it also builds upon Press et al., 2021), but in this case the modification of attention weights is fixed rather than learned. In contrast to the narrative by Ghosh et al. (2020), the more complex attention mechanism used in their model is *not* strictly exponentially decaying, since the proposed bias factor has a multiplicative rather than additive effect on attention scores $e_{t,t-\tau}$. While the importance of more recent interactions is also increased in this attention mechanism, the approach in general results in *set attention* for interactions farther in the past, due to attention scores decaying towards zero rather than minus infinity (as is the case in our approach).

### 4.3 SET REPRESENTATIONS OF INTERACTIONS

The elements of every interaction $\mathbf{y}_t = (\mathbf{q}_t, \mathbf{c}(\mathbf{q}_t), \mathbf{r}_t)$ are embedded separately and aggregated to form joint representations that are used as inputs to the transformer architecture. We require the aggregation function for interaction tuples $\mathbf{y}_t$ (and question tuples $\mathbf{x}_t$) to be *permutation invariant* (cf. Zaheer et al., 2017) to the ordering of knowledge components $\mathbf{c}(\mathbf{q}_t)$ associated with question $\mathbf{q}_t$. Permutation invariance is a desirable property of functions operating on sets, whose implementations

induce an ordered representation of its input. Formally, consider a question $\mathbf{q}$ with $k$ associated knowledge components and let $\mathbf{n} = \mathbf{n}(\mathbf{q})$ be their corresponding indices. Let $\pi$ denote a permutation of a $k$-tuple of integers 1 through $k$. For the aggregation function $\phi(\mathbf{x})$ to be permutation invariant with respect to the ordering of knowledge components, it must hold that

$$\phi(\mathbf{q}, \{c_{\mathbf{n}_1}, c_{\mathbf{n}_2}, \ldots, c_{\mathbf{n}_k}\}) = \phi(\mathbf{q}, \{c_{\mathbf{n}_{\pi(1)}}, c_{\mathbf{n}_{\pi(2)}}, \ldots, c_{\mathbf{n}_{\pi(k)}}\})$$

for any permutation $\pi$. This definition also applies to the ordering of knowledge components in $\phi(\mathbf{y})$.

Consider the following three permutation invariant aggregation functions: Firstly, the *mean* operation satisfies permutation invariance (cf. Zaheer et al., 2017). Secondly, a function that assigns an integer to any unique set is also permutation invariant (for illustration, consider that the function orders the set first and only then assigns an integer to the ordered tuple, in which case ordering yields permutation invariance). Thirdly, unmasked multi-head self-attention with standard scaled dot-product attention applied to a set of tokens without positional encoding yields permutation invariant representations per token, as attention weights are indifferent to the ordering of tokens (this property is usually referred to as permutation *equivariance*; for a formal proof compare for example Girgis et al., 2022).

Let $\mathbf{e}_* \in \mathbb{R}^d$ denote embedding vectors and let $\oplus$ denote element-wise addition. We propose three interaction embeddings $\mathbf{e_y}$ that all adhere to permutation invariant aggregation. Question embeddings $\mathbf{e_x}$ are computed analogously, but without response embeddings $\mathbf{e_r}$.

**Mean embeddings**  Firstly, we assign unique embedding vectors $\mathbf{e}_c$ to each knowledge component in $\mathbf{c}(\mathbf{q})$ and compute their mean. Final interaction embeddings are the result of element-wise addition with question $\mathbf{e_q}$ and response embedding $\mathbf{e_r}$:

$$\mathbf{e_y}^{\text{mean}} = \mathbf{e_q} \oplus \text{mean}(\{\mathbf{e}_c | c \in \mathbf{c}(\mathbf{q})\}) \oplus \mathbf{e_r} \tag{1}$$

**Unique set embeddings**  Secondly, we assign a unique embedding vector $\mathbf{e}_{\text{unique}(\{c|c\in\mathbf{c}(\mathbf{q})\})}$ to each unique set of knowledge components. The final interaction embedding is computed as above:

$$\mathbf{e_y}^{\text{unique}} = \mathbf{e_q} \oplus \mathbf{e}_{\text{unique}(\{c|c\in\mathbf{c}(\mathbf{q})\})} \oplus \mathbf{e_r}. \tag{2}$$

**MHSA embeddings**  Thirdly, we pass a sequence containing a *query token*, the question embedding $\mathbf{e_q}$ and embedding vectors $\mathbf{e}_c$ for knowledge components in $\mathbf{c}(\mathbf{q})$ through (possibly multiple layers) of unmasked multi-head self-attention $\text{MHSA}(\cdot)$. The final interaction embedding equals the element-wise addition of the transformed query token and the response embedding $\mathbf{e_r}$:

$$\mathbf{e_y} = \text{MHSA}(\{\text{query}\} \cup \{\mathbf{e_q}\} \cup \{\mathbf{e}_c | c \in \mathbf{c}(\mathbf{q})\})_{\text{query}} \oplus \mathbf{e_r} \tag{3}$$

All three proposed embeddings are straightforward and do not require any domain-knowledge. Mean embeddings (1) and unique set embeddings (2) have been used in prior studies on knowledge tracing (e.g. Long et al., 2021; Gervet et al., 2020, respectively). They are straightforward to implement and do not add significant computational cost. MHSA embeddings for knowledge tracing (3) are novel and come with high modeling capacity, however they do add computational cost.

Mean embeddings allow the models to attribute student performance to individual knowledge components, while interaction effects appear to be more difficult to learn. Unique set embeddings naturally account for interactions. However, we argue that it becomes increasingly difficult to attribute responses to individual knowledge components, as the approach results in a (theoretically) exponential increase in the number of embeddings, each of which has only a few occurrences. In principle, MHSA embeddings involve the most general and powerful aggregation of knowledge components, accounting for both individual and interaction effects. Empirically, we observe that for knowledge tracing tasks with small knowledge-component-to-question ratios, simple aggregations like mean embeddings and unique set embeddings of knowledge components seem to work equally well, while MHSA embeddings are difficult to optimize and result in overfitting. We conjecture, that MHSA embeddings perform best in more complicated settings with large data, as we see comparatively better performance for larger datasets with many knowledge components, such as the Ednet dataset (see Section 5.1). This conjecture is also supported by experiments on synthetic data, where we experiment with varying knowledge-component-to-question ratios and dataset sizes (see Section 5.3).

**Expanded and domain-inspired interaction representations in related work**    The interaction embeddings proposed for KTSTs build upon a principled set representation, use standard machine learning paradigms for feature extraction and are domain-agnostic. This is in contrast with many models proposed in related work. We highlight problems of the *expanded representation* of knowledge tracing sequences prevalent in recent publications (e.g. Ghosh et al., 2020; Liu et al., 2023b; 2022) in Section 3. Additionally, the expanded representation violates permutation invariance, as the order of knowledge components influences both the transformation within the model as well as final probability scores. The order of knowledge components is provided implicitly through causal masking or/and explicitly in the custom attention mechanism (e.g. Ghosh et al., 2020) or via positional encodings (e.g. Liu et al., 2023b). Furthermore, interaction representations proposed in related work are often domain inspired and unnecessarily complex. Specifically, we consider all three interaction embeddings proposed for KTSTs conceptually simpler than domain inspired *Rasch embeddings* used in Ghosh et al. (2020) and other recent knowledge tracing models. *Rasch embeddings* prescribe, that questions are represented as $\mathbf{e_x} = \mathbf{e}_c \oplus (\mathbf{e}_q \cdot \mathbf{v}_c)$, the addition of two knowledge component embeddings $\mathbf{e}_c \in \mathbb{R}^d$ and $\mathbf{v}_c \in \mathbb{R}^d$, where the latter is supposed to capture knowledge component specific variations and is scaled by $\mathbf{e}_q \in \mathbb{R}$, a scalar embedding of questions that "controls how far this question deviates from the [knowledge component] it covers" (Ghosh et al., 2020). Interactions are represented analogously with knowledge-component-response embeddings and knowledge-component-response variation vectors. Given these design choices, *Rasch embeddings* require the flawed expanded representation. In the next section, we provide empirical evidence, that straightforward set representations are sufficient to capture relevant domain information within knowledge tracing tasks.

## 5    EXPERIMENTS

In this section, we empirically evaluate the proposed knowledge tracing set transformers on a benchmark with standardized tasks, preprocessing, data splits, and fixed test sets for various educational datasets (*pykt*, Liu et al., 2022). We thus provide evidence that KTSTs set a new state-of-the-art. Furthermore, we verify our design choices within the transformer architecture (Section 4.1) as well as the proposed learnable modification of attention matrices used in KTSTs (Section 4.2) in an ablation study. We additionally experiment with synthetic data, generated according to classic multidimensional item response theory (MIRT, Reckase, 2009), to demonstrate the advantages of the proposed MHSA aggregation (Section 4.3) in large data knowledge tracing settings with high knowledge-component-to-question-ratios.

### 5.1    STATE-OF-THE-ART BENCHMARK RESULTS

We evaluate KTSTs on the *pykt* benchmark (Liu et al., 2022). In accordance with standard practice in knowledge tracing we report AUC and accuracy values. In summary, we experiment on eight publicly available datasets: Ednet, Algebra2005 (AL2005), ASSISTments2009 (AS2009), NeurIPS34, Bridge2006 (BD2006), Statics2011, ASSISTments2015 (AS2015), and POJ;[2] and we compare against the following baselines: DKT (Piech et al., 2015), DKVMN (Zhang et al., 2017), DKT+ (Yeung & Yeung, 2018), DeepIRT (Yeung, 2019), DKT-F (Nagatani et al., 2019), GKT (Nakagawa et al., 2019), KQN (Lee & Yeung, 2019), SAKT (Pandey & Karypis, 2019), SKVMN (Abdelrahman & Wang, 2019), AKT (Ghosh et al., 2020), qDKT (Sonkar et al., 2020), SAINT (Choi et al., 2020), ATKT (Guo et al., 2021), HawkesKT (Wang et al., 2021), IEKT (Long et al., 2021), LPKT (Shen et al., 2021), DIMKT (Shen et al., 2022), AT-DKT (Liu et al., 2023a), DTransformer (Yin et al., 2023). FoLiBiKT (Im et al., 2023), QIKT (Chen et al., 2023), and simpleKT (Liu et al., 2023b). We discuss notable baselines in related work in Section 2, in the problem setting in Section 3 as well as in contrast to our contribution in Section 4.

Models are trained on sequences of at most 200 consecutive interactions each and tested on entire interaction sequences of students in the test set.[3] All baseline results are reproduced by us with model implementations and hyperparameters provided by Liu et al. (2022). During hyperparameter

---

[2]We refer to Liu et al. (2022) for a description of the data, licenses, and detailed experimental setup.

[3]We only consider a sliding window of interactions with at most 200 knowledge components as learning history. For proper evaluation, we fixed a bug in the *pykt* evaluation framework (Liu et al., 2022) that is related to the *expanded representation*, where the restriction to a sequence length of 200 was equally applied to either *questions* for set based models and to *knowledge components* for models using the *expanded representation*.

Table 1: Benchmark AUC results. Markers $*$, $\circ$ and $\bullet$ indicate whether KTST (mean) is statistically superior, equal or inferior to baselines, respectively, using a paired $t$-test at the 0.01 significance level.

| | Ednet | AL2005 | AS2009 | NIPS34 | BD2006 |
|---|---|---|---|---|---|
| DKT | $0.6108 \pm 0.0017$ $*$ | $0.8137 \pm 0.0018$ $*$ | $0.7532 \pm 0.0012$ $*$ | $0.7682 \pm 0.0006$ $*$ | $0.8011 \pm 0.0005$ $*$ |
| DKVMN | $0.6158 \pm 0.0020$ $*$ | $0.8060 \pm 0.0016$ $*$ | $0.7461 \pm 0.0010$ $*$ | $0.7675 \pm 0.0004$ $*$ | $0.7980 \pm 0.0014$ $*$ |
| DKT+ | $0.6156 \pm 0.0018$ $*$ | $0.8142 \pm 0.0004$ $*$ | $0.7536 \pm 0.0016$ $*$ | $0.7688 \pm 0.0002$ $*$ | $0.8012 \pm 0.0006$ $*$ |
| GKT | $0.6213 \pm 0.0024$ $*$ | $0.8085 \pm 0.0018$ $*$ | $0.7422 \pm 0.0028$ $*$ | $0.7650 \pm 0.0071$ $*$ | $0.8043 \pm 0.0013$ $*$ |
| SAKT | $0.6074 \pm 0.0013$ $*$ | $0.7887 \pm 0.0042$ $*$ | $0.7245 \pm 0.0009$ $*$ | $0.7507 \pm 0.0011$ $*$ | $0.7732 \pm 0.0010$ $*$ |
| SKVMN | $0.6230 \pm 0.0045$ $*$ | $0.7461 \pm 0.0033$ $*$ | $0.7326 \pm 0.0016$ $*$ | $0.7502 \pm 0.0012$ $*$ | $0.7286 \pm 0.0046$ $*$ |
| AKT | $0.6705 \pm 0.0024$ $*$ | $0.8298 \pm 0.0017$ $*$ | $0.7840 \pm 0.0016$ $*$ | $0.8030 \pm 0.0003$ $*$ | $\underline{0.8204} \pm 0.0006$ $*$ |
| SAINT | $0.6598 \pm 0.0023$ $*$ | $0.7767 \pm 0.0018$ $*$ | $0.6918 \pm 0.0036$ $*$ | $0.7866 \pm 0.0023$ $*$ | $0.7762 \pm 0.0025$ $*$ |
| HawkesKT | $0.6815 \pm 0.0041$ $*$ | $0.8207 \pm 0.0021$ $*$ | $0.7224 \pm 0.0006$ $*$ | $0.7757 \pm 0.0014$ $*$ | $0.8067 \pm 0.0011$ $*$ |
| IEKT | $0.7301 \pm 0.0012$ $*$ | $0.8403 \pm 0.0019$ $*$ | $0.7832 \pm 0.0021$ $*$ | $0.8039 \pm 0.0003$ $*$ | $0.8116 \pm 0.0013$ $*$ |
| LPKT | $0.7340 \pm 0.0007$ $*$ | $0.8239 \pm 0.0008$ $*$ | $0.7811 \pm 0.0019$ $*$ | $0.7940 \pm 0.0012$ $*$ | $0.8039 \pm 0.0004$ $*$ |
| DIMKT | $0.6748 \pm 0.0030$ $*$ | $0.8276 \pm 0.0007$ $*$ | $0.7717 \pm 0.0010$ $*$ | $0.8022 \pm 0.0009$ $*$ | $0.8166 \pm 0.0007$ $*$ |
| DTransformer | $0.6719 \pm 0.0037$ $*$ | $0.8189 \pm 0.0024$ $*$ | $0.7718 \pm 0.0021$ $*$ | $0.7990 \pm 0.0006$ $*$ | $0.8083 \pm 0.0006$ $*$ |
| FoLiBiKT | $0.6721 \pm 0.0018$ $*$ | $0.8307 \pm 0.0005$ $*$ | $0.7828 \pm 0.0016$ $*$ | $0.8028 \pm 0.0005$ $*$ | $0.8203 \pm 0.0015$ $*$ |
| QIKT | $0.7260 \pm 0.0013$ $*$ | $0.8408 \pm 0.0008$ $*$ | $0.7877 \pm 0.0019$ $*$ | $\underline{0.8041} \pm 0.0008$ $*$ | $0.8094 \pm 0.0008$ $*$ |
| simpleKT | $0.6593 \pm 0.0041$ $*$ | $0.8246 \pm 0.0012$ $*$ | $0.7745 \pm 0.0021$ $*$ | $0.8035 \pm 0.0002$ $*$ | $0.8159 \pm 0.0005$ $*$ |
| KTST (mean) | $\mathbf{0.7394} \pm 0.0002$ | $\underline{0.8522} \pm 0.0004$ | $\mathbf{0.7993} \pm 0.0012$ | $\mathbf{0.8071} \pm 0.0000$ | $\mathbf{0.8264} \pm 0.0004$ |
| KTST (unique) | $0.7355 \pm 0.0008$ $*$ | $\mathbf{0.8529} \pm 0.0009$ $\circ$ | $\underline{0.7989} \pm 0.0014$ $\circ$ | — | — |
| KTST (MHSA) | $\underline{0.7389} \pm 0.0009$ $\circ$ | $0.8288 \pm 0.0009$ $*$ | $0.7871 \pm 0.0022$ $*$ | — | — |

Table 2: Benchmark accuracy results. Markers $*$, $\circ$ and $\bullet$ indicate whether KTST (mean) is statistically superior, equal or inferior to baselines, respectively, using a paired $t$-test at the 0.01 significance level.

| | Ednet | AL2005 | AS2009 | NIPS34 | BD2006 |
|---|---|---|---|---|---|
| DKT | $0.6420 \pm 0.0023$ $*$ | $0.8094 \pm 0.0010$ $*$ | $0.7241 \pm 0.0012$ $*$ | $0.7024 \pm 0.0008$ $*$ | $0.8551 \pm 0.0003$ $*$ |
| DKVMN | $0.6446 \pm 0.0030$ $*$ | $0.8031 \pm 0.0008$ $*$ | $0.7194 \pm 0.0006$ $*$ | $0.7020 \pm 0.0004$ $*$ | $0.8545 \pm 0.0003$ $*$ |
| DKT+ | $0.6517 \pm 0.0057$ $*$ | $0.8093 \pm 0.0004$ $*$ | $0.7241 \pm 0.0013$ $*$ | $0.7034 \pm 0.0005$ $*$ | $0.8551 \pm 0.0002$ $*$ |
| GKT | $0.6639 \pm 0.0050$ $*$ | $0.8086 \pm 0.0008$ $*$ | $0.7158 \pm 0.0016$ $*$ | $0.6956 \pm 0.0103$ $*$ | $0.8553 \pm 0.0003$ $*$ |
| SAKT | $0.6392 \pm 0.0039$ $*$ | $0.7959 \pm 0.0018$ $*$ | $0.7071 \pm 0.0016$ $*$ | $0.6870 \pm 0.0010$ $*$ | $0.8456 \pm 0.0006$ $*$ |
| SKVMN | $0.6606 \pm 0.0090$ $*$ | $0.7821 \pm 0.0032$ $*$ | $0.7160 \pm 0.0010$ $*$ | $0.6874 \pm 0.0010$ $*$ | $0.8408 \pm 0.0005$ $*$ |
| AKT | $0.6645 \pm 0.0035$ $*$ | $0.8125 \pm 0.0016$ $*$ | $0.7383 \pm 0.0020$ $*$ | $0.7317 \pm 0.0005$ $*$ | $\underline{0.8586} \pm 0.0005$ $*$ |
| SAINT | $0.6511 \pm 0.0039$ $*$ | $0.7789 \pm 0.0030$ $*$ | $0.6885 \pm 0.0044$ $*$ | $0.7172 \pm 0.0025$ $*$ | $0.8374 \pm 0.0108$ $\circ$ |
| HawkesKT | $0.6905 \pm 0.0025$ $*$ | $0.8112 \pm 0.0012$ $*$ | $0.7045 \pm 0.0008$ $*$ | $0.7102 \pm 0.0013$ $*$ | $0.8559 \pm 0.0005$ $*$ |
| IEKT | $0.7106 \pm 0.0018$ $*$ | $0.8228 \pm 0.0008$ $*$ | $0.7336 \pm 0.0027$ $*$ | $0.7327 \pm 0.0001$ $*$ | $0.8556 \pm 0.0009$ $*$ |
| LPKT | $0.7128 \pm 0.0004$ $*$ | $0.8129 \pm 0.0008$ $*$ | $0.7356 \pm 0.0011$ $*$ | $0.7179 \pm 0.0033$ $*$ | $0.8538 \pm 0.0002$ $*$ |
| DIMKT | $0.6700 \pm 0.0038$ $*$ | $0.8106 \pm 0.0004$ $*$ | $0.7354 \pm 0.0019$ $*$ | $0.7309 \pm 0.0005$ $*$ | $0.8578 \pm 0.0004$ $*$ |
| DTransformer | $0.6656 \pm 0.0032$ $*$ | $0.8054 \pm 0.0007$ $*$ | $0.7284 \pm 0.0007$ $*$ | $0.7290 \pm 0.0012$ $*$ | $0.8556 \pm 0.0006$ $*$ |
| FoLiBiKT | $0.6666 \pm 0.0028$ $*$ | $0.8127 \pm 0.0012$ $*$ | $0.7391 \pm 0.0013$ $*$ | $0.7319 \pm 0.0005$ $*$ | $0.8583 \pm 0.0006$ $*$ |
| QIKT | $0.7077 \pm 0.0014$ $*$ | $0.8220 \pm 0.0007$ $*$ | $0.7382 \pm 0.0008$ $*$ | $0.7326 \pm 0.0008$ $*$ | $0.8537 \pm 0.0005$ $*$ |
| simpleKT | $0.6565 \pm 0.0029$ $*$ | $0.8081 \pm 0.0010$ $*$ | $0.7319 \pm 0.0019$ $*$ | $\underline{0.7327} \pm 0.0003$ $*$ | $0.8579 \pm 0.0002$ $*$ |
| KTST (mean) | $\mathbf{0.7154} \pm 0.0011$ | $\underline{0.8287} \pm 0.0005$ | $\mathbf{0.7490} \pm 0.0013$ | $\mathbf{0.7356} \pm 0.0003$ | $\mathbf{0.8608} \pm 0.0005$ |
| KTST (unique) | $0.7131 \pm 0.0017$ $\circ$ | $\mathbf{0.8291} \pm 0.0008$ $\circ$ | $\underline{0.7489} \pm 0.0011$ $\circ$ | — | — |
| KTST (MHSA) | $\underline{0.7152} \pm 0.0011$ $\circ$ | $0.8166 \pm 0.0012$ $*$ | $0.7415 \pm 0.0014$ $*$ | — | — |

optimization, a budget of 200 runs has been granted for each combination of baseline and data fold. We refer to Liu et al. (2022) for more details on search spaces and tuning procedure. Hyperparameters of KTST models are tuned according to a tree-structured Parzen estimator (Bergstra et al., 2011; Akiba et al., 2019), with a budget of 100 runs for each data fold. Model selection is performed via a 5-fold cross validation using AUC as criterion. Details on the hyperparameter optimization can be found in Appendix A.1.

Table 1 (AUC) and 2 (accuracy) show mean performance and standard deviations for KTSTs, for most datasets and most baselines. We refer to complete results in Appendix A.3 which include experiments regarding datasets Statics2011, AS2015, and POJ and baselines DeepIRT, DKT-F, KQN, qDKT, ATKT, and AT-DKT. KTST (mean), KTST (unique) and KTST (MHSA) refer to KTST architectures with mean embeddings, unique set embeddings, and MHSA embeddings, respectively. For datasets with a knowledge-component-to-question ratio of 1.0, or approximately 1.0, we do not provide KTST (unique) and KTST (MHSA) results as they are expected to match the results with mean embeddings. KTST models achieve state-of-the-art AUC results on all datasets except Statics2011. For accuracy the results are similar. Paired $t$-tests further show that almost all improvements over baselines are significant at a significance level of 0.01.

Table 3: Ablation study on AS2009. Markers $*$ and $\circ$ indicate whether learnable ALiBi (q=k) is statistically superior or equal, respectively, using a paired $t$-test at the 0.01 significance level.

| Attention mechanism | AUC | ACC |
|---|---|---|
| Standard MHA + PE (q$\neq$k) | $0.7744 \pm 0.0014$ $*$ | $0.7307 \pm 0.0015$ $*$ |
| AKT (q=k) | $0.7958 \pm 0.0011$ $*$ | $0.7464 \pm 0.0009$ $*$ |
| ALiBi (q$\neq$k) | $0.7953 \pm 0.0018$ $*$ | $0.7456 \pm 0.0021$ $\circ$ |
| ALiBi (q=k) | $\underline{0.7978} \pm 0.0010$ $*$ | $\underline{0.7479} \pm 0.0007$ $\circ$ |
| Learnable ALiBi (q$\neq$k) | $0.7976 \pm 0.0013$ $\circ$ | $0.7476 \pm 0.0012$ $\circ$ |
| Learnable ALiBi (q=k) | $\mathbf{0.7993} \pm 0.0012$ | $\mathbf{0.7490} \pm 0.0013$ |
| Learnable ALiBi (q=k) decoder-only | $0.7977 \pm 0.0011$ $*$ | $0.7473 \pm 0.0007$ $\circ$ |

The principled and permutation invariant aggregation of knowledge components is one of the strengths of KTSTs. Compared to baselines using the flawed *expanded representation*, we thus expect the largest gains in performance for the datasets Ednet, AL2005 and AS2009 with an average knowledge-component-per-question ratio of $2.30$, $1.46$, and $1.19$, respectively. The results confirm our hypothesis with differences being most pronounced on Ednet, where IEKT, LPKT, and QIKT are the only baselines that are in the same ballpark as KTST results. Notably, all three models are also based on set representations. KTST (mean) generally performs well. Unique set embeddings turn out to be more suited for small component-to-question ratios, as they seem to incur a penalty for higher knowledge-component-to-question ratios, whereas KTST (MHSA)'s performance is only competitive on Ednet. We conjecture that KTST (MHSA)'s modeling capacity might be too high for simpler knowledge tracing tasks. We support this conjecture in experiments on synthetic data in Section 5.3.

## 5.2 ABLATION OF ARCHITECTURE AND ATTENTION MECHANISM

Table 3 provides the results of an ablation study conducted on the AS2009 dataset, where we compare KTST (mean) with four different types of attention mechanisms: *Standard MHA + PE* refers to standard multi-head attention with positional embeddings, *AKT* refers to the attention mechanism proposed in Ghosh et al. (2020), *ALiBi* refers to the attention mechanism proposed in Press et al. (2021) for language models, and *Learnable ALiBi* refers to the learnable modification of attention matrices that we propose to use in KTSTs. Additionally, entries $q = k$ and $q \neq k$ refer to whether the *query* is set to equal the *key* in the cross-attention. In KTSTs, we set $q = k$ following related work (notably Pandey & Karypis, 2019). *Decoder-only* refers to an architecture without an encoder (corresponding to the KTST architecture with number of encoding layers set to 0) as suggested in Zhan et al. (2024). We report the mean and standard deviation of test results based on training with 5-fold validation. As can be seen, *Learnable ALiBi* with *query* set equal to the *key* and an encoder-decoder architecture performs best, confirmed by a paired $t$-tests at a significance level of 0.01. This supports our design choices for KTSTs and explains observed performance gains in the benchmark setting to some extend.

## 5.3 CAPACITY OF PROPOSED SET REPRESENTATIONS

In this section, we report on synthetic data generated according to classic multidimensional item response theory (Reckase, 2009), where we train on sequences with varying number of knowledge components per questions. We thereby investigate the effect of proposed aggregation methods for set representations of interactions within KTSTs. Specifically, we sample interactions from a compensatory multidimensional 3PL model (Reckase, 2009). For question $i$ and learner $j$, the probability of a correct response is given by

$$P\left(\mathbf{r} = 1 | a_i, b_i, c_i\right) = c_i + \frac{(1 - c_i)}{1 + \exp\left(a_i^\top (\theta_j - b_i)\right)},$$

where $c_i$ denotes the probability of guessing a correct response for question $i$, $b_i \in \mathbb{R}^k$ and $\theta_j \in \mathbb{R}^k$ are vectors with latent difficulties and student skills per knowledge component, respectively. A randomly sampled multi-hot vector $a_i \in \{0, 1\}^k$ assigns knowledge components to questions, with $k$ knowledge components in total. We set $c_i = 0.25$ for all questions and sample both $b_i$ and $\theta_i$ from

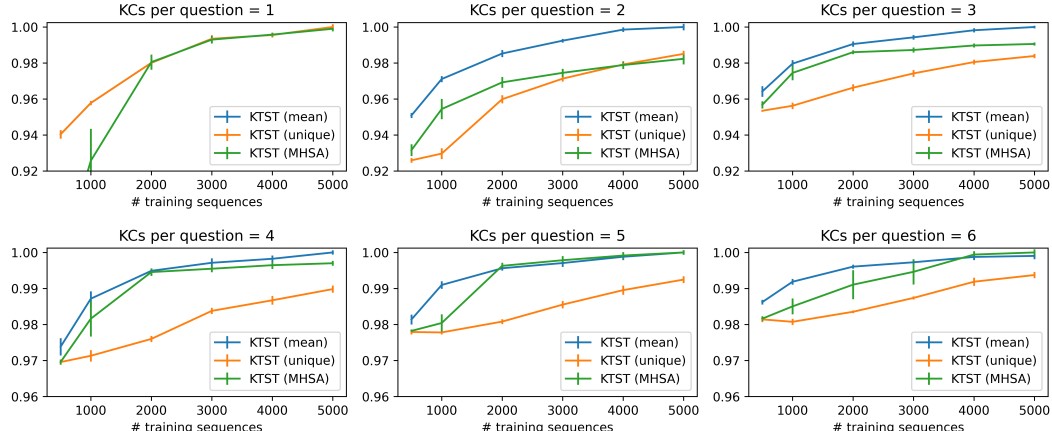

Figure 3: Relative AUC performance of KTST models for synthetic MIRT data with different embedding aggregations and varying numbers of knowledge components (KCs) per question. The highest AUC is normalized to $1.00$ for each experiment; error bars indicate 1-sigma standard error.

a multivariate normal distribution. If a student interacts with a knowledge component, we increase her latent skill. Setting the number of questions to 1,000 and the total number of components to 10, we simulate $40$ interactions per student and train KTST models on different sample sizes. For each configuration we train 5 models and report test results on 1,000 interaction sequences. Figure 3 visualizes results of the experiment.

As expected, mean embeddings and unique set embeddings achieve the same results on a problem with only one knowledge component per question, while mean embeddings outperform unique set embeddings in settings with more than one knowledge component per question (the line for KTST (mean) is hidden by the line for KTST (unique) in the upper left plot of Figure 3). We conjecture that this relates to the (in theory) exponentially increasing number of unique combinations of set elements. MHSA embeddings perform relatively poor in settings with little data and a low knowledge-component-to-question-ratio, but show the best performance in more complicated settings with larger sample size and more knowledge components per question. This is in line with our conjecture that KTST models using MHSA embeddings have too much capacity for some of the simpler real world benchmark datasets, which in turn would render optimization difficult.

## 6 CONCLUSION

In this paper, we proposed knowledge tracing set transformers (KTSTs), a straightforward, yet principled and performant model class for knowledge tracing prediction tasks. We proposed a simplified variant of learnable modification of attention matrices, as well as principled set representations of student interactions that are permutation invariant with respect to sets of knowledge components and that do not rely on domain knowledge. As a result, KTSTs are conceptually simpler than previous state-of-the-art approaches. We proposed three different interaction representations that come with different properties. The simplest representation, based on mean embeddings, empirically performed best. In contrast, the representation with the highest capacity, based on MHSA embeddings, showed promising results for more involved settings with large sample sizes and multiple knowledge components per question. Overall, KTSTs establish new state-of-the-art performance for knowledge tracing prediction tasks.

A limitation of KTSTs is that the model class does not include an interpretable internal state that reflects the current knowledge state of students (cf. Gervet et al., 2020). However, a qualitative inspection of the implicit knowledge representation could be insightful, for example by using post-hoc model-agnostic interpretations (e.g. Rodrigues et al., 2022). This could be a valuable avenue in future work.

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

## A APPENDIX

### A.1 REPRODUCIBILITY

For the submission to ICLR 2025, we provide a zip-File of the codebase for KTSTs, including an implementation of proposed models as well as training and evaluation setup. The code includes everything required to reproduce our experiments and results and comes with appropriate instructions. In case of acceptance, we will extend this Section with more detailed information and instructions and publish our code via a publicly available code repository. Hyperparameter sweeps for KTST models requires roughly 1200 hours of GPU time. Training only the best hyperparameter configurations for all of our experiments is considerably cheaper.

### A.2 OPTIMIZATION DETAILS

**Learnable ALiBi** ALiBi (Press et al., 2021), the attention mechanism we build upon in KTSTs, was originally introduced for improved extrapolation to longer sequence lengths in language models. In experiments conducted by Press et al. (2021), a learnable ALiBi variant was not found to be helpful for NLP tasks. In contrast, we find that learning decay parameters $\theta$ (as introduced in Section 4.2) comes with a statistically significant performance improvement for KTSTs (cf. Section 5.2). We however noticed that optimization required a higher learning rate for $\theta$ compared to other model parameters. Details can be found in configuration files for the benchmark experiments and ablation.

**Regularization of question embeddings** Empirical results for KTSTs suggest, that strong regularization of question embeddings $\mathbf{e_q}$ is important for high performance on knowledge tracing tasks. In practice, we propose to initialize all parameters in question embeddings to zero, by setting $\mathbf{e_q} = \mathbf{0}$. In an ablation study, we have retrained the best model configurations of KTST (mean) for the experiments on the benchmark datasets in Section 5.1 without this *0-init* initialization. Results are provided in Tables 4 and 5, where we observe a significant drop in performance on most datasets (markers $*$ and $\circ$ indicate whether KTST (mean) is statistically superior or equal, respectively, using a paired *t*-test at the 0.01 significance level). Overall, the ablation study thus supports our choice of initialization. Question embedding initialization was previously identified as important for the more complex *Rasch embeddings* (within AKT, by Ghosh et al., 2020, compare also our description of Rasch embeddings in Section 4.3). Notably, we find that some recent publications do not properly account for initialization within baseline models (cf. Lee et al., 2022; Im et al., 2023).

Table 4: AUC results for ablation of question embedding initialization within KTST (mean)

|  | Ednet | AL2005 | AS2009 | NIPS34 | BD2006 |
|---|---|---|---|---|---|
| **KTST (w/o 0-init)** | $0.7251 \pm 0.0032$ $*$ | $0.8425 \pm 0.0006$ $*$ | $0.7762 \pm 0.0010$ $*$ | $0.8067 \pm 0.0003$ $\circ$ | $0.8142 \pm 0.0008$ $*$ |
| **KTST** | $0.7394 \pm 0.0002$ | $0.8522 \pm 0.0004$ | $0.7993 \pm 0.0012$ | $0.8071 \pm 0.0000$ | $0.8264 \pm 0.0004$ |

Table 5: Accuracy results for ablation of question embedding initialization within KTST (mean)

|  | Ednet | AL2005 | AS2009 | NIPS34 | BD2006 |
|---|---|---|---|---|---|
| **KTST (w/o 0-init)** | $0.7092 \pm 0.0013$ $*$ | $0.8239 \pm 0.0008$ $*$ | $0.7346 \pm 0.0012$ $*$ | $0.7352 \pm 0.0007$ $\circ$ | $0.8572 \pm 0.0003$ $*$ |
| **KTST** | $0.7154 \pm 0.0011$ | $0.8287 \pm 0.0005$ | $0.7490 \pm 0.0013$ | $0.7356 \pm 0.0003$ | $0.8608 \pm 0.0005$ |

### A.3 COMPLETE RESULTS FOR BENCHMARK EXPERIMENTS

In Tables 6 and 7 we provide complete results for benchmark experiments as described in Section 5.1. Specifically, we add results for models DeepIRT (Yeung, 2019), DKT-F (Nagatani et al., 2019), KQN (Lee & Yeung, 2019), qDKT (Sonkar et al., 2020), ATKT (Guo et al., 2021), and AT-DKT (Liu et al., 2023a) as well as datasets Statics2011, ASSISTments2015 (AS2015), and POJ. All our claims hold.

Table 6: Complete benchmark AUC results. Markers *, ∘ and ● indicate whether KTST (mean) is statistically superior, equal or inferior to baselines, respectively.

| | Ednet | AL2005 | AS2009 | NIPS34 | BD2006 | Statics2011 | AS2015 | POJ |
|---|---|---|---|---|---|---|---|---|
| **DKT** | 0.6108 ± 0.0017 * | 0.8137 ± 0.0018 * | 0.7532 ± 0.0012 * | 0.7682 ± 0.0006 * | 0.8011 ± 0.0005 * | 0.8220 ± 0.0015 * | 0.7268 ± 0.0005 * | 0.6092 ± 0.0010 * |
| **DKVMN** | 0.6158 ± 0.0020 * | 0.8060 ± 0.0016 * | 0.7461 ± 0.0010 * | 0.7675 ± 0.0004 * | 0.7980 ± 0.0014 * | 0.8092 ± 0.0016 * | 0.7223 ± 0.0004 * | 0.6056 ± 0.0022 * |
| **DKT+** | 0.6156 ± 0.0018 * | 0.8142 ± 0.0004 * | 0.7536 ± 0.0016 * | 0.7688 ± 0.0002 * | 0.8012 ± 0.0006 * | 0.8275 ± 0.0004 ∘ | 0.7284 ± 0.0006 * | 0.6173 ± 0.0007 * |
| **DeepIRT** | 0.6171 ± 0.0024 * | 0.8030 ± 0.0014 * | 0.7459 ± 0.0006 * | 0.7668 ± 0.0008 * | 0.7961 ± 0.0008 * | 0.8048 ± 0.0040 * | 0.7217 ± 0.0003 ∘ | 0.6040 ± 0.0019 * |
| **DKT-F** | 0.6164 ± 0.0008 * | 0.8147 ± 0.0016 * | — | 0.7729 ± 0.0003 * | 0.7979 ± 0.0014 * | 0.7805 ± 0.0008 * | — | 0.6027 ± 0.0026 * |
| **GKT** | 0.6213 ± 0.0024 * | 0.8085 ± 0.0018 * | 0.7422 ± 0.0028 * | 0.7650 ± 0.0071 * | 0.8043 ± 0.0013 * | 0.8033 ± 0.0063 * | 0.7233 ± 0.0015 * | 0.6051 ± 0.0064 * |
| **KQN** | 0.6086 ± 0.0022 * | 0.8023 ± 0.0021 * | 0.7465 ± 0.0015 * | 0.7672 ± 0.0004 * | 0.7932 ± 0.0008 * | 0.8231 ± 0.0007 * | 0.7255 ± 0.0004 * | 0.6079 ± 0.0015 * |
| **SAKT** | 0.6074 ± 0.0013 * | 0.7887 ± 0.0042 * | 0.7245 ± 0.0009 * | 0.7507 ± 0.0011 * | 0.7732 ± 0.0010 * | 0.7961 ± 0.0013 * | 0.7117 ± 0.0002 * | 0.6091 ± 0.0013 * |
| **SKVMN** | 0.6230 ± 0.0045 * | 0.7461 ± 0.0033 * | 0.7326 ± 0.0016 * | 0.7502 ± 0.0012 * | 0.7286 ± 0.0046 * | 0.8071 ± 0.0030 * | 0.7076 ± 0.0006 * | 0.5996 ± 0.0023 * |
| **AKT** | 0.6705 ± 0.0024 * | 0.8298 ± 0.0017 * | 0.7840 ± 0.0016 * | 0.8030 ± 0.0003 * | 0.8204 ± 0.0006 * | **0.8308 ± 0.0012** ∘ | 0.7279 ± 0.0008 * | 0.6281 ± 0.0015 * |
| **qDKT** | 0.6986 ± 0.0006 * | 0.7480 ± 0.0014 * | 0.7014 ± 0.0053 * | 0.7998 ± 0.0003 * | 0.7521 ± 0.0007 * | — | — | — |
| **SAINT** | 0.6598 ± 0.0023 * | 0.7767 ± 0.0018 * | 0.6918 ± 0.0036 * | 0.7866 ± 0.0023 * | 0.7762 ± 0.0025 * | 0.7602 ± 0.0129 * | 0.7015 ± 0.0009 * | 0.5564 ± 0.0013 * |
| **ATKT** | 0.6048 ± 0.0019 * | 0.7975 ± 0.0007 * | 0.7453 ± 0.0005 * | 0.7657 ± 0.0005 * | 0.7871 ± 0.0017 * | 0.8049 ± 0.0023 * | 0.7238 ± 0.0008 * | 0.6071 ± 0.0015 * |
| **HawkesKT** | 0.6815 ± 0.0041 * | 0.8207 ± 0.0021 * | 0.7224 ± 0.0006 * | 0.7757 ± 0.0014 * | 0.8067 ± 0.0011 * | — | — | — |
| **IEKT** | 0.7301 ± 0.0012 * | 0.8403 ± 0.0019 * | 0.7832 ± 0.0021 * | 0.8039 ± 0.0003 * | 0.8116 ± 0.0013 * | — | — | — |
| **LPKT** | 0.7340 ± 0.0007 * | 0.8239 ± 0.0008 * | 0.7811 ± 0.0019 * | 0.7940 ± 0.0012 * | 0.8039 ± 0.0004 * | — | — | — |
| **DIMKT** | 0.6748 ± 0.0030 * | 0.8276 ± 0.0007 * | 0.7717 ± 0.0010 * | 0.8022 ± 0.0009 * | 0.8166 ± 0.0007 * | — | — | — |
| **AT-DKT** | 0.6207 ± 0.0047 * | 0.8223 ± 0.0027 * | 0.7550 ± 0.0017 * | 0.7813 ± 0.0004 * | 0.8086 ± 0.0011 * | — | — | — |
| **DTransformer** | 0.6719 ± 0.0037 * | 0.8189 ± 0.0024 * | 0.7718 ± 0.0021 * | 0.7990 ± 0.0006 * | 0.8083 ± 0.0006 * | 0.8235 ± 0.0020 * | 0.7257 ± 0.0004 * | 0.6176 ± 0.0009 * |
| **FoLiBiKT** | 0.6721 ± 0.0018 * | 0.8307 ± 0.0005 * | 0.7828 ± 0.0016 * | 0.8028 ± 0.0005 * | 0.8203 ± 0.0015 * | 0.8302 ± 0.0010 ∘ | 0.7283 ± 0.0004 * | 0.6283 ± 0.0006 * |
| **QIKT** | 0.7260 ± 0.0013 * | 0.8408 ± 0.0008 * | 0.7877 ± 0.0019 * | 0.8041 ± 0.0008 * | 0.8094 ± 0.0008 * | — | — | — |
| **simpleKT** | 0.6593 ± 0.0041 * | 0.8246 ± 0.0012 * | 0.7745 ± 0.0021 * | 0.8035 ± 0.0002 * | 0.8159 ± 0.0005 * | 0.8192 ± 0.0003 * | 0.7245 ± 0.0006 * | 0.6248 ± 0.0009 * |
| **KTST (mean)** | **0.7394 ± 0.0002** | 0.8522 ± 0.0004 | **0.7993 ± 0.0012** | **0.8071 ± 0.0000** | **0.8264 ± 0.0004** | 0.8291 ± 0.0009 | **0.7314 ± 0.0003** | **0.6347 ± 0.0011** |
| **KTST (unique)** | 0.7355 ± 0.0008 * | **0.8529 ± 0.0009** ∘ | 0.7989 ± 0.0014 ∘ | — | — | — | — | — |
| **KTST (MHSA)** | 0.7389 ± 0.0009 ∘ | 0.8288 ± 0.0009 * | 0.7871 ± 0.0022 * | — | — | — | — | — |

Table 7: Complete benchmark accuracy results. Markers $*$, $\circ$ and $\bullet$ indicate whether KTST (mean) is statistically superior, equal or inferior to baselines, respectively.

| | Ednet | AL2005 | AS2009 | NIPS34 | BD2006 | Statics2011 | AS2015 | POJ |
|---|---|---|---|---|---|---|---|---|
| **DKT** | $0.6420 \pm 0.0023$ * | $0.8094 \pm 0.0010$ * | $0.7241 \pm 0.0012$ * | $0.7024 \pm 0.0008$ * | $0.8551 \pm 0.0003$ * | $0.7974 \pm 0.0005$ * | $0.7505 \pm 0.0006$ * | $0.6329 \pm 0.0023$ * |
| **DKVMN** | $0.6446 \pm 0.0030$ * | $0.8031 \pm 0.0008$ * | $0.7194 \pm 0.0006$ * | $0.7020 \pm 0.0004$ * | $0.8545 \pm 0.0003$ * | $0.7931 \pm 0.0008$ * | $0.7507 \pm 0.0003$ * | $0.6394 \pm 0.0016$ * |
| **DKT+** | $0.6517 \pm 0.0057$ * | $0.8093 \pm 0.0004$ * | $0.7241 \pm 0.0013$ * | $0.7034 \pm 0.0005$ * | $0.8551 \pm 0.0002$ * | $0.7974 \pm 0.0006$ $\circ$ | $0.7509 \pm 0.0004$ * | $0.6479 \pm 0.0023$ * |
| **DeepIRT** | $0.6483 \pm 0.0058$ * | $0.8033 \pm 0.0007$ * | $0.7191 \pm 0.0007$ * | $0.7014 \pm 0.0006$ * | $0.8542 \pm 0.0001$ * | $0.7909 \pm 0.0030$ * | $0.7507 \pm 0.0002$ * | $0.6372 \pm 0.0008$ * |
| **DKT-F** | $0.6395 \pm 0.0009$ * | $0.8090 \pm 0.0006$ * | — | $0.7070 \pm 0.0004$ * | $0.8534 \pm 0.0006$ * | $0.7867 \pm 0.0006$ * | — | $0.6371 \pm 0.0038$ * |
| **GKT** | $0.6639 \pm 0.0050$ * | $0.8086 \pm 0.0008$ * | $0.7158 \pm 0.0016$ * | $0.6956 \pm 0.0103$ * | $0.8553 \pm 0.0003$ * | $0.7900 \pm 0.0011$ * | $0.7496 \pm 0.0006$ * | $0.6024 \pm 0.0228$ * |
| **KQN** | $0.6387 \pm 0.0031$ * | $0.8023 \pm 0.0013$ * | $0.7224 \pm 0.0014$ * | $0.7018 \pm 0.0002$ * | $0.8533 \pm 0.0004$ * | $0.7981 \pm 0.0008$ $\circ$ | $0.7502 \pm 0.0002$ * | $0.6433 \pm 0.0020$ * |
| **SAKT** | $0.6392 \pm 0.0039$ * | $0.7959 \pm 0.0018$ * | $0.7071 \pm 0.0016$ * | $0.6870 \pm 0.0010$ * | $0.8456 \pm 0.0006$ * | $0.7877 \pm 0.0021$ * | $0.7474 \pm 0.0001$ * | $0.6399 \pm 0.0029$ * |
| **SKVMN** | $0.6606 \pm 0.0090$ * | $0.7821 \pm 0.0032$ * | $0.7160 \pm 0.0010$ * | $0.6874 \pm 0.0010$ * | $0.8408 \pm 0.0005$ * | $0.7922 \pm 0.0010$ * | $0.7457 \pm 0.0004$ * | $0.6407 \pm 0.0029$ * |
| **AKT** | $0.6645 \pm 0.0035$ * | $0.8125 \pm 0.0016$ * | $0.7383 \pm 0.0020$ * | $0.7317 \pm 0.0005$ * | $\underline{0.8586} \pm 0.0005$ * | $\mathbf{0.8023} \pm 0.0006$ $\circ$ | $0.7521 \pm 0.0005$ $\circ$ | $0.6492 \pm 0.0017$ * |
| **qDKT** | $0.6922 \pm 0.0005$ * | $0.7262 \pm 0.0008$ * | $0.6781 \pm 0.0034$ * | $0.7305 \pm 0.0003$ * | $0.8302 \pm 0.0007$ * | | | |
| **SAINT** | $0.6511 \pm 0.0039$ * | $0.7789 \pm 0.0030$ * | $0.6885 \pm 0.0044$ * | $0.7172 \pm 0.0025$ * | $0.8374 \pm 0.0108$ * | $0.7630 \pm 0.0139$ * | $0.7451 \pm 0.0008$ * | $0.6474 \pm 0.0005$ * |
| **ATKT** | $0.6364 \pm 0.0015$ * | $0.7988 \pm 0.0008$ * | $0.7201 \pm 0.0009$ * | $0.7007 \pm 0.0006$ * | $0.8507 \pm 0.0003$ * | $0.7905 \pm 0.0016$ * | $0.7490 \pm 0.0003$ * | $0.6364 \pm 0.0032$ * |
| **HawkesKT** | $0.6905 \pm 0.0025$ * | $0.8112 \pm 0.0012$ * | $0.7045 \pm 0.0008$ * | $0.7102 \pm 0.0013$ * | $0.8559 \pm 0.0005$ * | | — | — |
| **IEKT** | $0.7106 \pm 0.0018$ * | $0.8228 \pm 0.0008$ * | $0.7336 \pm 0.0027$ * | $0.7327 \pm 0.0001$ * | $0.8556 \pm 0.0009$ * | | — | — |
| **LPKT** | $0.7128 \pm 0.0004$ * | $0.8129 \pm 0.0008$ * | $0.7356 \pm 0.0011$ * | $0.7179 \pm 0.0033$ * | $0.8538 \pm 0.0002$ * | | — | — |
| **DIMKT** | $0.6700 \pm 0.0038$ * | $0.8106 \pm 0.0004$ * | $0.7354 \pm 0.0019$ * | $0.7309 \pm 0.0005$ * | $0.8578 \pm 0.0004$ * | | — | — |
| **AT-DKT** | $0.6474 \pm 0.0066$ * | $0.8130 \pm 0.0009$ * | $0.7246 \pm 0.0016$ * | $0.7146 \pm 0.0006$ * | $0.8553 \pm 0.0004$ * | | — | — |
| **DTransformer** | $0.6656 \pm 0.0032$ * | $0.8054 \pm 0.0007$ * | $0.7284 \pm 0.0007$ * | $0.7290 \pm 0.0012$ * | $0.8556 \pm 0.0006$ * | $0.7988 \pm 0.0021$ $\circ$ | $0.7511 \pm 0.0006$ $\circ$ | $0.6509 \pm 0.0004$ * |
| **FoLiBiKT** | $0.6666 \pm 0.0028$ * | $0.8127 \pm 0.0012$ * | $0.7391 \pm 0.0013$ * | $0.7319 \pm 0.0005$ * | $0.8583 \pm 0.0006$ * | $\underline{0.8019} \pm 0.0005$ $\circ$ | $\mathbf{0.7526} \pm 0.0002$ $\circ$ | $0.6507 \pm 0.0033$ $\circ$ |
| **QIKT** | $0.7077 \pm 0.0014$ * | $0.8220 \pm 0.0007$ * | $0.7382 \pm 0.0008$ * | $0.7326 \pm 0.0008$ * | $0.8537 \pm 0.0005$ * | | — | — |
| **simpleKT** | $0.6565 \pm 0.0029$ * | $0.8081 \pm 0.0010$ * | $0.7319 \pm 0.0019$ * | $\underline{0.7327} \pm 0.0003$ * | $0.8579 \pm 0.0002$ * | $0.7947 \pm 0.0026$ * | $0.7505 \pm 0.0005$ * | $\underline{0.6516} \pm 0.0009$ * |
| **KTST (mean)** | $\mathbf{0.7154} \pm 0.0011$ | $\underline{0.8287} \pm 0.0005$ | $\mathbf{0.7490} \pm 0.0013$ | $\mathbf{0.7356} \pm 0.0003$ | $\mathbf{0.8608} \pm 0.0005$ | $0.8002 \pm 0.0014$ | $0.7523 \pm 0.0002$ | $\mathbf{0.6568} \pm 0.0008$ |
| **KTST (unique)** | $0.7131 \pm 0.0017$ $\circ$ | $\mathbf{0.8291} \pm 0.0008$ $\circ$ | $\underline{0.7489} \pm 0.0011$ $\circ$ | — | — | — | — | — |
| **KTST (MHSA)** | $\underline{0.7152} \pm 0.0011$ $\circ$ | $0.8166 \pm 0.0012$ * | $0.7415 \pm 0.0014$ * | — | — | — | — | — |