# OpenReview forum: "Toward Principled Transformers for Knowledge Tracing"
_ICLR.cc/2025/Conference — ICLR 2025 Conference Withdrawn Submission_

### Official Review · Reviewer_gSxz · 2024-10-29

**Soundness:** 2
**Presentation:** 3
**Contribution:** 1
**Rating:** 3
**Confidence:** 4

**Summary:**

The paper introduces Knowledge Tracing Set Transformers (KTSTs), a simpler, principled model class for knowledge tracing tasks that avoids complex, domain-specific designs by employing set representations of student interactions and a learnable attention modification for positional information.

**Strengths:**

- This paper focuses on knowledge tracing, which is an important and interesting topic in the educational community.

- The writing is clear and the structure easy to follow.

- The authors compare a substantial number of baselines, which enhances the credibility of the experimental results.

**Weaknesses:**

- The novelty of this paper may not fully meet the conference’s expectations. It appears to incorporate established methods to enhance model performance without sufficiently clarifying the research problem and motivation. The paper claims to improve ALiBi by making its matrix parameter learnable to provide positional information. However, a similar approach for learnable ALiBi matrix parameter was already proposed in [1]. How does this paper’s method differ from that approach?

- Although the authors propose ALiBi with learnable parameters to supply position information to attention and introduce aggregation functions, the ablation study focuses solely on ALiBi, which appears insufficiently comprehensive. It remains unclear to what extent each component affects model performance.

- Figure 3 shows little differences among various aggregation functions. It's unclear why simulated data is used instead of real data.

- The paper claims the proposed method is “simpler than previous state-of-the-art approaches,” yet does not provide relevant experiments, such as an analysis of parameter count or computational cost.

- Although significance testing is conducted, the performance improvement observed is modest. From a practical standpoint, it remains uncertain whether this advancement is substantial enough to significantly impact the field of knowledge tracing.

- Figure 2 lacks clarity and omits essential annotations. For instance, what are the specific inputs for Q, K, and V? What does the pink box in the lower left corner represent?

- There is no analysis of the learnable attention matrices to investigate what exactly influences model performance. I recommend adding experiments to enhance understanding.


---
[1] Chi, Ta-Chung, et al. "Kerple: Kernelized relative positional embedding for length extrapolation." NeurIPS, 2022.

**Questions:**

- What are the differences between the improved ALiBi in this paper and the method in [1]?

- What is the motivation for using ALiBi in knowledge tracing models?

- What are the main contributions this paper?

- Why is simulated data used instead of real data in Section 5.3?

---

### Official Review · Reviewer_eo5z · 2024-10-31

**Soundness:** 2
**Presentation:** 2
**Contribution:** 1
**Rating:** 3
**Confidence:** 4

**Summary:**

The paper proposes Knowledge Tracing Set Transformers (KTSTs) for predicting student performance. Unlike domain-specific models, KTSTs use Transformer as a backbone and a learnable attention mechanism to handle student interaction data. KTSTs also learn set representations for knowledge components. The model outperforms or matches state-of-the-art results on multiple educational benchmarks.

**Strengths:**

The paper is clearly presented, effectively clarifying both the motivation and model architecture. The Transformer-based model is also straightforward to understand and implement. Additionally, the evaluations are quite comprehensive, comparing performance across 22 benchmark models, which clearly demonstrates the advantages of KTST in predicting student performance.

**Weaknesses:**

- This work contrasts itself with models incorporating domain-specific knowledge (i.e., models with more interpretable components). However, the literature review overlooks a lot of newly published interpretable models. The focus of the review should shift towards domain-inspired and other Transformer-based models rather than general deep learning models. The authors should expand the review to include recent interpretable models and provide a comparative analysis.

- In Sections 2 and 4, the repeated claim that “In contrast to KTSTs, most related work includes domain-inspired components that increase model complexity” lacks a clear comparison of these components and a quantification of the added complexity. Thus, I have the following questions and confusions:

  1. I am unclear on the criticism of existing domain-inspired models
      - The inductive biases in architectures discussed in the paper, such as memory-augmented neural networks, question difficulty models, and graph neural networks for knowledge structures, are well-motivated within the educational domain. They reflect human learning processes, are generally beneficial in educational contexts and are not so specific as to be limited to particular subjects or cultures. From my perspective, the contribution of this work—specifically, the multi-head attention mechanism with learnable exponential decay on attention weights—is a re-formulation of embedding memory priors into the model. Similarly, the permutation invariance of concept representations functions as another form of regularization on the concept graph structure.

      - Regarding “interaction representations proposed in related work are often domain inspired and unnecessarily complex. “ Could the authors provide concrete examples on the domain-inspired and unnecessarily complex embeddings in existing works?

  2. For the compared benchmarks, could the authors compare the complexity of these models quantitatively, including the training and evaluation time and the amount of model parameters?

  3. Domain-inspired models are generally motivated by their effectiveness with small datasets and their interpretability, rather than purely optimizing prediction performance. To evaluate this, I suggest two additional experiments:

      - Could the authors conduct experiments on smaller datasets? Currently, models are trained on sequences of up to 200 consecutive interactions, which is extensive for educational data. Reducing the sequence length and the number of students would provide insight into model performance on limited data.

      - Could the authors analyze the embeddings learned by the model, such as the representations for knowledge components and question embeddings? This would provide interpretability insights into the regularization of permutation invariance.

- I do appreciate the insight that concept representations should be permutation invariant. Could the authors include an ablation study to examine the impact of this design choice? Specifically: 1) test the model without enforcing permutation invariance on concept representations; 2) remove the knowledge component embeddings altogether and only keep the question embeddings.

- I find it challenging to pinpoint the technical contributions of this work. The learnable attention weights component (Section 4.2) seems primarily to add flexibility to existing domain-inspired representations. Additionally, although three choices for set representations are explored, the novel approach (MHSA) offers only marginal advantages over mean embedding when training sequences exceed 4,000 interactions and each question includes 6 knowledge components in synthetic data. This seems not that applicable to real-world datasets as shown in the experiments on KT data.

**Questions:**

Please see weaknesses.

---

### Official Review · Reviewer_aeYy · 2024-11-03

**Soundness:** 2
**Presentation:** 1
**Contribution:** 2
**Rating:** 1
**Confidence:** 2

**Summary:**

The paper tackles the task of knowledge tracing, which can sort of be summarized to predicting a binary correctness value for a response R to a question Q with certain knowledge components C (i.e concepts), conditioned on the previous questions (and their components and responses). The authors use the transformer architecture for this (which seem to also be explored in prior work) and propose a specific Multi-head Self-Attention block that is suited to the task. Specifically, it is argued that the the Knowledge Components should be parsed in a permutation-agnostic way. There are also other approaches such as simply taking the mean of all the components that are studied.

The experiments study quantitatively study such design choices and also explore other aspects of the data, such as the average number of components per question and how it relates to the used method.

**Strengths:**

**(a)** The related work discussion is very thorough and, even though I'm not familiar with the literature, the authors do a great job in covering many relevant works on every single design choice. The presentation of such related work, however, needs some rework (see weakness **(a)** below)

**(b)** The experiments seem to be very thorough (it is mentioned that there are multiple initialization and hyper-parameters tested for every baseline). The ablation is also well-done in terms of different design choices (cf. Table 3. with inputting different queries to the attention blocks or Table 2&3 (bottom) with different component aggregation strategies).

**(c)** The numbers overall also seem promising. Although I'm confused whether the (mean) and (unique) lines from Tables 2 and 3 can be considered as their contribution? (see Weakness **(d)** below).

**Weaknesses:**

**(a)** I really appreciate the authors being thorough when discussing the related work, but I think there is a lot of room for improving the discussion. Specifically, the current version can sometimes be very confusing to someone who is not entirely familiar with the literature.

For example, take a look at Lines 108-136 (‘Limitations of related work’). I don’t understand why this section has to come after the Problem Setting (Section 3) and why it is not already covered in the previous Related Work discussion (Section 2). I understand that the authors needed to rely on the defined notation (in Line 121), but most of the text could be merged in the previous discussion on the Related Work. You could also potentially move the problem setting to precede Section 2.

The related work discussion is again further spread to other sections, see the Lines 198-207 and Lines 252-259. This really makes it hard for me to assess where this paper stands in comparison to related work, as suddenly later it is revealed that there are other works who have also explored similar directions, which didn’t seem to be the case in Section 2. I urge the authors to significantly rework the related work discussion. For example, you could include a paragraph on all the transformer-based Knowledge Tracing methods and maybe also a paragraph on the works who explored different attention mechanisms. These should all be in the Section 2 (Related Work) and not interleaved in the method discussion!

One other example of this issue would be Line 286-288: “We propose three interaction embeddings”. And later in Line 307-208 two of which (`mean` and `unique`) turn out to be used in prior work and only the third one is claimed novel. Again, as a reader, it is very confusing where this paper stands.

**(b)** The overall related work discussion can also sometimes be too abstract. For example, Lines 127-128 state “Without proper masking …, this introduces label leakage”. This sentence for example is not clear what is meant by neither “proper masking” nor “label leakage”.


**(c)**  There are certain instances where the authors criticize prior work on weak grounds. For example, in Lines 200-202 it is stated that prior RNN-based work has certain inductive biases by having a hidden state associated with a student's knowledge, and later it is stated that the paper’s [transformer-based] approach is conceptually simpler. I cannot 100% agree with such a comparison. I don’t agree why having an inductive bias, as long as it’s general-enough, would be a downside and certainly don’t agree with transformers being “conceptually simpler” than an RNN.

**(d)** Regarding the results, it seems that the most-relevant line in the tables is `KTST (MHSA)` as it is stated in line 309-310  that the `(mean)` and `(unique)` methods are from prior work. If that is the case, then the tables actually don't seem that promising since it's always under-performing prior work? I would like to ask the authors to clarify this.


----- Minor Issues -----

**(e)** The very first paragraph in the introduction needs 1-2 more sentences to further clarify what the setting is. It should clarify that the context is student-computer interaction and this is supposed to be used for digital education.

**(f)** Line 122-123, at the end, it is stated: “One consequence is an increase …”. But later in the text there is no second consequence.

**(g)** Currently Figure 2 is not very optimal, as it is basically the vanilla transformer architecture, except for X, Y, and Rs as inputs and outputs. I think the figure should be with further detail, demonstrating all the token types (e.g knowledge components and questions). A significant part of your method is also the aggregation, which again the figure is not explaining. The “causal masking” should ideally also be demonstrated in the figure.

**(h)** I think the Figure 1 is really not that informative. It's just with random numbers and shapes and it really doesn't describe the problem. I think something like the figures in this talk (https://youtu.be/8ITtYnhslvE?si=ExSW6WGShqNTTTiu&t=106) would be more informative to someone not familiar with this topic.

**Questions:**

**(a)** Regarding the expanded representation works, I wanted to ask the authors how it works for a query with more than one knowledge component. Specifically, let’s say for a query X_5, after passing all the interactions Y_1 to Y_4, how do we query X_5 if it has more than one knowledge component? If for example the query has 5 knowledge components, do we have to give 4 queries (with each knowledge component in a single query) and simply ignore the output until the last (5th) query is given?

**(b)** Regarding the results, Lines 515-520, state that MHSA works better for larger data and larger Component-to-question ratio. But at the same time MHSA (compared to the mean method), introduces learnable parameters that require data. So I’m not 100% sure if the issue is really the ratio, or just the low amount of data in certain datasets? I would appreciate it if this can be disentangled (with an experiment).

---

> ### Comment · Reviewer_aeYy · 2024-11-26
> **Lowering the score**
>
> Unfortunately the authors did not engage in any discussion regarding the reviews (neither mine or others'). Therefore I would decrease my score from Reject to Strong Reject.

---

### Official Review · Reviewer_b2fc · 2024-11-04

**Soundness:** 2
**Presentation:** 3
**Contribution:** 2
**Rating:** 5
**Confidence:** 4

**Summary:**

This paper presents knowledge tracing set transformers (KTSTs), a class of streamlined models designed specifically for knowledge tracing prediction tasks. To account for the unique characteristics of these tasks, this work introduces a simplified, learnable variant of the attention matrix and an interaction representation that does not rely on domain-specific knowledge. In experiments on standardized benchmark datasets, KTST achieves new state-of-the-art performance.

**Strengths:**

1.KTST demonstrates promising performance.
2.The authors conducted comprehensive comparative experiments and provided an in-depth analysis of the results.
3.The model diagram is clear and straightforward, enhancing readability and understanding.

**Weaknesses:**

1. The primary motivation is not clearly presented in the text, and the structure of sections reveals some issues in logical flow, with parts containing redundant explanations.
2. The expression “This model class … flawed evaluation” in the abstract is convoluted and unclear, making it difficult for readers to grasp the core motivation of the study.
3. The Introduction lacks an illustrative figure that directly presents the research problem, and the content in this section appears insufficient.
4. While I understand that the authors place part of the Related Work in Section 4 to emphasize their contributions, the extensive descriptions might raise questions regarding the sufficiency of the work’s original contributions.
5. Section 4.2 primarily introduces the learnable modification of attention matrices. Could you please explain how it differs from ALiBi [1]?
6. Section 4.3 mainly addresses the handling of multi-concept and identifies it as one of the paper’s research questions. As far as I know, related works [2,3,4] have largely resolved this issue, so what are the significant advantages of this approach?
7. The experiments in Section 5.3 introduce a randomly simulated dataset for multi-concept knowledge, aimed at comparing three embedding methods. In addition to random simulations, including a specially designed simulation method could make the comparisons more compelling.

[1] Press O, Smith N A, Lewis M. Train short, test long: Attention with linear biases enables input length extrapolation[J]. arXiv preprint arXiv:2108.12409, 2021.

[2] Long T, Liu Y, Shen J, et al. Tracing knowledge state with individual cognition and acquisition estimation[C]//Proceedings of the 44th International ACM SIGIR Conference on Research and Development in Information Retrieval. 2021: 173-182.

[3] Zhang M, Zhu X, Zhang C, et al. Multi-factors aware dual-attentional knowledge tracing[C]//Proceedings of the 30th ACM International Conference on Information & Knowledge Management. 2021: 2588-2597.

[4] Cui J, Chen Z, Zhou A, et al. Fine-grained interaction modeling with multi-relational transformer for knowledge tracing[J]. ACM Transactions on Information Systems, 2023, 41(4): 1-26.

**Questions:**

Refer to the weakness above.

---

### Author Response · Authors · 2024-11-20

Dear PC,

The main arguments against the paper seem to deal with the perceived lack of technical novelty. By contrast, our paper does not primarily aim to introduce novel technical contributions. Instead, the paper aims to set things straight in an important interdisciplinary domain. We show that many of the previously introduced concepts used by state-of-the-art approaches are unnecessary for performance, lead to overly complex models, introduce distribution shift between training and testing, and have previously led to mistakes in empirical evaluation. Our contribution addresses the KT prediction problem with on par or better performance but in a technically straightforward, correct, and simple way.

Sincerely yours,
the authors

---

### Note · Authors · 2024-11-26

**Comment:**

Dear PC,

We are withdrawing the paper.
We thank all reviewers for their time and for their comments.

Sincerly yours,
the authors

**Withdrawal Confirmation:**

I have read and agree with the venue's withdrawal policy on behalf of myself and my co-authors.